# A Theoretical and Practical Analysis of Membrane Protein Genes Altered in Neutrophils in Parkinson’s Disease

**DOI:** 10.3390/cimb47060459

**Published:** 2025-06-13

**Authors:** Araliz López Pintor, Miriam Nolasco López, José Daniel Lozada-Ramírez, Martín Alejandro Serrano-Meneses, Alicia Ortega Aguilar, Dante Oropeza Canto, César Flores-de los Ángeles, Victor Hugo Anaya-Muñoz, Aura Matilde Jiménez-Garduño

**Affiliations:** 1Department of Chemical and Biological Sciences, Universidad de las Américas Puebla, Cholula 72810, Mexico; araliz.lopezpr@udlap.mx (A.L.P.); miriam.nolascolz@udlap.mx (M.N.L.); jose.lozada@udlap.mx (J.D.L.-R.); martin.serrano@udlap.mx (M.A.S.-M.); 2Biochemistry Department, Medical Faculty, Universidad Nacional Autónoma de México, Mexico City 04510, Mexico; aortega@unam.mx; 3Neurosciences Clinic, Hospital Ángeles Puebla, Puebla 72190, Mexico; danteoropeza2@hotmail.com; 4Molecular Diagnostics Laboratory, Integral Health Center, Universidad Popular Autónoma del estado de Puebla, Puebla 72410, Mexico; cesar.flores@upaep.mx; 5Escuela Nacional de Estudios Superiores, Unidad Morelia, Universidad Nacional Autónoma de México, Morelia 58190, Mexico; 6Health Sciences Department, Universidad de las Américas Puebla, Cholula 72810, Mexico

**Keywords:** neutrophil, Parkinson’s disease, membrane protein, gene expression and digital PCR

## Abstract

Parkinson’s disease (PD) is a major health concern, with no accurate or early diagnostic test available for most patients. Chronic inflammation is a recognized contributor to PD pathogenesis; thus, membrane proteins of inflammatory cells such as neutrophils present an accessible target for detecting early molecular changes. In this study, we conducted a theoretical analysis using the GSE99039 database to identify differentially expressed genes (DEGs) in leukocytes from PD patients. From this, we selected nine top candidates for digital polymerase chain reaction (dPCR) analysis in isolated neutrophils from nine PD patients and nine matched controls. Our results revealed significant upregulation of *ORAI3* and *CLCN2*. Unexpectedly, both *ACTB* (*β*-actin) and *SNCA* (alpha-synuclein) were also upregulated in neutrophils. Notably, this study provides the first evidence of *CLCN2* expression in neutrophils and demonstrates the significant upregulation of four genes via dPCR. These genes may serve as potential biomarkers for future research on PD detection.

## 1. Introduction

Parkinson’s disease (PD) is the second most common neurodegenerative disorder [1,2,3], characterized by both motor and non-motor symptoms. Non-motor symptoms often emerge during the earliest stages of the disease but are frequently overlooked due to their mild presentation and overlap with common age-related conditions such as anosmia, pain, depression, anxiety, sleep disturbances, and cognitive impairment [4]. As the disease progresses over several years, the degeneration of dopaminergic neurons in the substantia nigra pars compacta leads to the onset of primary motor symptoms, including resting tremor, bradykinesia, rigidity, postural instability, and electroencephalographic abnormalities—manifestations that may vary by sex and age [5,6,7,8].

The main pathophysiological hallmark of motor symptoms is the aggregation of intraneuronal alpha-synuclein (α-syn) in structures known as Lewy bodies (LBs) [9]. While the etiology of PD is not fully understood [10], approximately 5–10% of cases have a genetic basis, with mutations in genes such as *SNCA* (encoding α-syn), *DJ-1*, *PINK1*, and *LRRK2* leading to early-onset forms of the disease; however, the majority of PD cases are idiopathic and age-associated [11,12]. In addition to genetic susceptibility, various environmental risk factors—including exposure to pesticides, heavy metals, traumatic brain injury, and bacterial or viral infections—have been implicated in PD and are strongly associated with inflammatory processes [13,14,15,16,17].

Multiple studies have demonstrated that neuroinflammation plays a pivotal role in PD pathophysiology and is closely linked to both its initiation and progression [18,19]. Notably, shared genetic variants have been identified between PD and autoimmune or inflammatory disorders such as Crohn’s disease, further underscoring the role of the immune system in PD pathogenesis [20,21]. Recent studies by Qu et al. reported elevated levels of proinflammatory cytokines in both the peripheral blood and cerebrospinal fluid of PD patients [22,23]. Postmortem analyses of brain tissue from PD patients have revealed the presence of activated microglia as well as infiltration by altered inflammatory and non-immune cells [24,25,26]. Beyond the well-documented microgliosis and astrogliosis observed in PD brains, peripheral inflammation and the influence of PD risk-associated genes contribute significantly to the chronic inflammatory environment that drives the progression of this neurodegenerative disorder [27].

In recent decades, neutrophils have attracted increasing attention due to their involvement in chronic inflammation. Their role in autoimmune and other chronic degenerative diseases is well established, and research into their contribution to neurodegenerative disorders is now emerging. Neutrophil-induced inflammatory changes have been reported in several neurodegenerative conditions [28,29]; however, their specific role in the pathophysiology of PD remains unclear, underscoring the need for further investigation. As the most abundant leukocytes and being easily accessible in peripheral blood, neutrophils represent promising candidates for identifying diagnostic or disease progression biomarkers—particularly through the study of their membrane proteins. These proteins are highly suitable for clinical applications due to their surface accessibility. Indeed, membrane proteins, including G protein-coupled receptors and ion channels, constitute approximately 51% of the molecular targets of currently approved drugs [30]. Consequently, these proteins were the focus of our investigation.

The primary aim of this study was to identify ion channels and immune receptors that are differentially expressed in leukocytes from PD patients using a publicly available transcriptomic database. Based on this analysis, we selected candidate genes of interest for digital polymerase chain reaction (dPCR) analysis using cDNA synthesized from neutrophil RNA isolated from both PD patients and healthy controls. Our results provide evidence of the differential regulation of neutrophil membrane proteins during the course of PD pathophysiology.

## 2. Materials and Methods

### 2.1. Selection of Microarray Database from Gene Expression Omnibus (GEO)

A microarray dataset for PD was obtained from the GEO (https://www.ncbi.nlm.nih.gov/geo/, accessed on April 2023) using the following search terms: (“genes”[MeSH Terms] OR “gene”[All Fields]) AND “idiopathic”[All Fields] AND (“parkinson disease”[MeSH Terms] OR “parkinson’s disease”[All Fields]) AND (“blood”[Subheading] OR “blood”[MeSH Terms] OR “blood”[All Fields]) AND (“humans”[MeSH Terms] OR “Homo sapiens”[Organism] OR “Homo sapiens”[All Fields]).

Given the lack of neutrophil-specific datasets, we selected the GSE99039 database, which contains gene expression data from whole blood samples of PD patients and healthy controls and includes more than 50 participants.

### 2.2. Microarray Data Processing

The GSE99039 database includes samples from 500 individuals, either diagnosed with idiopathic PD or serving as healthy controls. Using the GEO2R tool (https://www.ncbi.nlm.nih.gov/geo/geo2r/ accessed on April 2023), we analyzed differentially expressed genes (DEGs) between 233 healthy controls and 205 PD patients. Data were downloaded and analyzed in April 2023. Forty-eight samples from individuals with other neurodegenerative diseases were excluded [31]. DEGs with *p* ≥ 0.05 were categorized as upregulated, while those with *p* ≤ 0.05 were considered downregulated.

### 2.3. Atlas Protein Review

The presence of ion channels, immune receptors, and cytokines related to neuroinflammation in granulocytes and/or neutrophils was confirmed using the Human Protein Atlas (https://www.proteinatlas.org). We included *CLCN2* despite the absence of reported expression in granulocytes due to its known associations with neurological disorders such as leukoencephalopathy, spinal and bulbar muscular atrophy (SBMA), and epilepsy [32,33,34].

### 2.4. Enrichment and Protein–Protein Network Analysis

Functional enrichment analysis was performed using ShinyGO v0.81 (http://bioinformatics.sdstate.edu/go/), which enabled the classification of genes by biological processes (BPs), cellular components (CCs), and molecular functions (MFs) [35]. Pathway analysis was conducted using the Kyoto Encyclopedia of Genes and Genomes (KEGG; https://www.kegg.jp) to assess gene function in the context of biological systems [36,37]. Global protein–protein interactions were evaluated using STRING (https://string-db.org). The combined information was compiled into an internal database [38].

### 2.5. Statistical Analyses

Data normality was assessed using the Shapiro–Wilk test. Student’s *t*-test was used for comparisons of normally distributed continuous variables; otherwise, the Mann–Whitney *U* test was applied. One-way ANOVA was used for comparisons between multiple normally distributed groups, followed by the Bonferroni post hoc test. For non-parametric group comparisons, the Kruskal–Wallis test was used. Results are presented as means ± standard deviations, and *p* < 0.05 was considered statistically significant. The false discovery rate (FDR) was calculated. All analyses were conducted using GraphPad Prism version 10 (https://www.graphpad.com).

### 2.6. Selection Criteria for Blood Samples

Nine PD patients were recruited by a neurology specialist at the Neurology Clinic, Hospital Ángeles Puebla (HAP). Peripheral blood samples were obtained after participants provided signed informed consent.

PD inclusion criteria: patients ≥ 60 years of age of either sex, diagnosed with idiopathic PD with at least 4 years of disease progression, and receiving anti-Parkinson’s treatment.

PD exclusion criteria: patients with neurological disorders unrelated to PD, acute or chronic inflammation not caused by PD, infectious diseases, autoimmune conditions, or those receiving anti-inflammatory treatments.

PD elimination criteria: coagulated samples and/or failure in processing.

The control group (CG) consisted of nine healthy volunteers meeting the following criteria: CG inclusion criteria: healthy individuals ≥60 years of age, of either sex, providing informed consent without compensation.CG exclusion criteria: subjects with infectious diseases, symptoms of chronic inflammation (unrelated to PD), autoimmune disorders, or anti-inflammatory treatment.CG elimination criteria: coagulated samples and/or processing failures.

### 2.7. Sample Collection

Peripheral blood samples (EDTA-anticoagulated) were collected from nine PD patients and nine controls. Samples were divided into aliquots for each experimental procedure.

### 2.8. Neutrophil Isolation

Neutrophils were isolated using PolymorphPrep™ (Seruwerk Bernburg AG, Oslo, Norway), following the manufacturer’s protocol.

### 2.9. RNA Extraction from Neutrophils

Total RNA from neutrophil-enriched fractions was extracted using the Total RNA Isolation Kit Tissue Column-Based (BIO-HELIX, Taiwan), according to the manufacturer’s instructions. RNA was stored at −80 °C and quantified using a NanoDrop spectrophotometer (Thermo Fisher Scientific, Waltham, MA, USA).

### 2.10. Transcription of mRNA to cDNA

cDNA was synthesized using RScript Reverse Transcriptase (BIO-HELIX Co., LTD., New Taipei City, Taiwan), quantified with a NanoDrop spectrophotometer, and stored at −80 °C.

### 2.11. End-Point PCR

End-point PCR was performed using a C1000 Touch Thermal Cycler (BIO-RAD, Hercules, CA, USA) and GoTaq^®^ 1-Step RT-qPCR System (Promega, Madison, WI, USA) with 10 ng of cDNA. Amplicons were separated on 2% agarose gels (NORGEN BIOTEK, Thorold, ON, Canada) using 1X TAE buffer (KARAL, England, UK) at 120 V for 60 min. Gels were imaged using the VisionWorks Acquisition and analysis version 8.20.17096.9551 software (Analytik, Jena, Germany). DNA fragment sizes were estimated using the BH 1 Kb Plus DNA Ladder RTU (BIO-HELIX Co., LTD., New Taipei City, Taiwan).

Gene sequences used for PCR included *FCGR3A* (NM_001127596.2), *PRG2* (NM_001243245.3), *FCGR2B*(NM_004001.5), *FCGR3B* (NM_001271037.2), and *CXCR2* (XM_047444187.1). Primers were designed using PerlPrimer, Primer-BLAST (NCBI), and SnapGene Sequences are provided in Appendix A.

### 2.12. Digital PCR

Digital PCR was performed using the QIAcuity Digital PCR System (QIAGEN, Hilden, Germany) with QIAcuity Nanoplates and the QIAcuity EG PCR Kit (QIAGEN, Hilden, Germany), using 10 ng of cDNA. Copy numbers were determined using the QIAcuity instrument and analyzed with the corresponding QIAcuity Software Suite 2.1.8.20..

Gene targets included *SNCA* (NG_011851), *ACTB* (NM_001101.5) *CACNG8* (NM_034895), *CLCN2* (NM_016422), *KCNE4* (NM_080671), *KCNJ15* (NM_170736), *ORAI3* (NM_152288), *P2RX1* (NG_012109), *TLR1* (NG_016228), and *TLR2* (NG_016229). Primers were designed using PerlPrimer and SnapGene. Final sequences are shown in Appendix A.

### 2.13. Ethical Considerations

This study was approved by the Research and Ethics Committees of Hospital Ángeles Puebla (Protocol Registration Number: CI-034-2022). Informed consent was obtained from all participants, and all personal data were kept confidential.

## 3. Results

The GSE99039 database comprises 486 gene expression profiles derived from whole blood samples. From this dataset, we identified 460 DEGs reported to be expressed in neutrophils or granulocytes, which were either upregulated or downregulated in PD patients compared to CGs. Among these, 279 genes encoded voltage-regulated ion channels, 110 were cell membrane receptor genes, and 71 were cytokines (Appendix A). Of the 463 DEGs analyzed, 34 were upregulated and 62 were downregulated in PD patients (Appendix A and Figure 1).

### 3.1. Selection of Main Genes in Granulocytes and/or Neutrophils Associated with PD

The 460 selected DEGs were analyzed using Student’s t-test to identify genes showing significant differences between the PD group and the CG. A total of 31 genes—including voltage-regulated ion channels, cytokines, and membrane receptors—were identified and are listed in Table 1 in ascending order of false discovery rate (FDR). Particular interest was placed on the *CLCN2* gene due to its known association with neurological disorders such as leukoencephalopathy and epilepsy, despite the fact that its expression has not previously been reported in granulocytes.

### 3.2. Enrichment Pathways Analysis

Functional enrichment analysis of the genes reported in Table 1 revealed that DEGs between PD and CG were predominantly involved in processes related to signal transduction, immune response, inflammatory response, and cytokine signaling (Figure 2 and Appendix A).

Figure 2 shows the fold enrichment for the top 30 pathways out of a total of 88 identified. As shown in Figure 2A, the highest enrichment levels—excluding those associated with microbial infections—were observed for the pathways “inflammatory bowel disease”, “Th1 and Th2 cell differentiation”, “Th17 cell differentiation”, “cytokine–cytokine receptor interaction”, “renin secretion”, and “JAK-STAT signaling pathway.” Although less enriched, pathways related to “neurodegeneration”, “Alzheimer’s disease”, and “neuroactive ligand–receptor interaction” also appeared, indicating that leukocyte (likely neutrophil) membrane proteins, including ion channels, are differentially expressed during PD.

In terms of the number of genes involved (Figure 2B), the most significant pathway was “cytokine–cytokine receptor interaction”, comprising 14 genes. This was followed by “inflammatory bowel disease” and “JAK-STAT signaling pathway”, each involving seven genes. Additionally, the pathways “Th1 and Th2 cell differentiation”, “Th17 cell differentiation”, “calcium signaling”, and “MAPK signaling” each included six genes. PD-related pathways such as “pathways of neurodegeneration—multiple diseases” and “Alzheimer’s disease” involved five and four genes, respectively. These altered gene expression patterns further support the need for deeper investigation into the role of these immune cells in the pathophysiology of PD.

### 3.3. Protein-Protein Interaction (PPI) Network Analysis

Our results indicate that the gene interaction network consists of 31 inflammation-related nodes (Figure 3). To identify the most interconnected genes within this network, we analyzed the ten genes with the highest node degrees. The genes exhibiting the highest connectivity were *IL1B* (degree = 18), *TNFRSF1A* (degree = 14), *IL4R* (degree = 13), *IL12RB1* (degree = 11), and *IL2RA* (degree = 10). Additional highly connected nodes included *IL10RB*, *IL17RA*, *IL18R1*, and *TLR2* (each with degree = 9), as well as *IL2RG* and *TNFRSF1B* (each with degree = 8). A higher node degree reflects greater interaction within the network [39]. 

### 3.4. Selection of Genes as Potential Biomarkers of PD

A total of nine genes were selected for further analysis via dPCR based on their network connectivity and statistical significance of expression. We intentionally excluded cytokines and cytokine receptors from this selection, as these functions are already extensively studied in neutrophils. Instead, our objective was to identify molecular targets that may offer novel insights into neutrophil roles during neurodegeneration.

Accordingly, we selected four genes encoding voltage-dependent ion channels (*CLCN2*, *CACNG8*, *KCNCE4*, and *KCNJ15*), four genes encoding membrane receptors (*TLR1*, *TLR2*, *ORAI3*, and *P2XR1*), and *SNCA* . We also included *ACTB* (β-actin) in the experiments. Figure 4 shows the statistically significant differences in gene expression (Log2-transformed fold changes) between groups identified in the GSE99039 dataset, along with the corresponding *p*-values.

### 3.5. Demographic Characteristics of PD and CG

The demographic characteristics of both groups are summarized in Table 2. The PD group consisted of nine individuals (six men and three women), with a mean age of 65.42 ± 9.75 years. Two participants in this group had a clinical diagnosis of high blood pressure (HBP). The mean duration of PD since diagnosis was 9.75 ± 6.38 years. The CG also included nine individuals (five men and four women), with a mean age of 63.92 ± 7.73 years and one subject diagnosed with HBP. No statistically significant differences were observed between the two groups for these demographic parameters.

### 3.6. Digital PCR Analysis

Complementary DNA was synthesized from neutrophil-enriched fractions obtained from 18 blood samples for dPCR analysis. Following neutrophil isolation, the enriched fractions were evaluated qualitatively through blood smear examination and endpoint PCR for markers of monocytes, basophils, and neutrophils, as shown in Appendix A.

Mean expression levels, reported in copies/µL, are presented in Table 3. The analysis identified four genes with significant differential expression between groups: *ACTB* (β-actin), *SNCA*, *CLCN2*, and *ORAI3*. All four genes were found to be upregulated in the PD group (Figure 5A–D). Notably, these findings are inconsistent with those from the theoretical analysis, where *SNCA* and *CLCN2* were reported to be downregulated in the PD group (Figure 4).

## 4. Discussions

Parkinson’s disease lacks standardized neuropathological benchmarks and reliable diagnostic tools applicable to premortem, accessible tissues. Given the growing evidence supporting the role of inflammation in PD, peripheral blood represents a valuable and accessible source for identifying potential biomarkers across different disease stages. The GSE99039 database provides comprehensive gene expression data from leukocytes of PD patients, enabling the identification of candidate biomarkers, as previously demonstrated by Jiang et al. (2019) [40]. Building on this resource, our group focused on the gene expression of neutrophil membrane proteins and, for the first time, demonstrated the expression and upregulation of an ion channel in neutrophils from PD patients.

The analysis of the 31 statistically significant DEGs from the GSE99039 database revealed, as expected, the regulation of several broad inflammatory pathways, including “Th1, Th2, and Th17 differentiation”, “JAK-STAT signaling”, and “cytokine–cytokine receptor interaction.” Interestingly, we also observed enrichment in pathways typically associated with nervous tissue—such as “neurodegeneration”, “Alzheimer’s disease”, and “neuroactive ligand–receptor interaction”—within leukocytes. We attribute these findings to the inclusion of ion channels and membrane receptors in our analysis. To our knowledge, this is the first theoretical evidence suggesting that previously unrecognized interactions between membrane proteins, particularly ion channels (Table 1), may play a role in neutrophils and contribute to neurodegenerative processes, including caspase activation, lipid peroxidation, mitochondrial dysfunction, and dysregulated autophagy (https://www.genome.jp/pathway/hsa05022; https://www.genome.jp/pathway/hsa05010 (https://www.genome.jp/kegg/pathway.html).

This hypothesis is further supported by the lack of known protein–protein interactions between ion channels and cytokine receptors, as revealed by our STRING analysis (Figure 3). For instance, *CLCN2* showed no interactions, *KCNH4* interacted only with *KCNH3*, and *ORAI1* and *ORAI3* interacted exclusively with each other. Calcium channel-related proteins (e.g., *CACNG8*, *CACNA1E*, and *CACN1S*) interacted among themselves and with potassium channels (*KCNJ15* and *KCNE4*) but not with other protein classes. Our results suggest that to better understand the enrichment of neurodegenerative pathways in leukocytes, future research must investigate potential interactions between ion channels, cytokine receptors, oxidative stress proteins, and calcium regulators—especially in neutrophils.

The second part of our study focused on nine DEGs selected based on statistical significance and biological interest, particularly ion channels in neutrophils. While recent studies have explored blood biomarkers in PD, most proposed candidates lack disease specificity. For example, a Pennsylvania cohort identified four serum proteins—bone sialoprotein (BSP), osteomodulin (OMD), aminoacylase-1 (ACY1), and growth hormone receptor (GHR)—as PD biomarkers, yet these were not specific to PD when compared to ALS or mild cognitive impairment [41]. Similarly, a study in Iowa found altered plasma cytokine levels in PD (increased IL-6 and IL-4; decreased IFNγ), which indicate inflammation but lack PD specificity [42]. A more recent single-cell RNA sequencing study in China highlighted the potential of the XCL2 receptor in NK cells as a biomarker [43] , reinforcing the importance of leukocyte research for diagnostic and prognostic purposes. Within this context, our results—identifying four differentially expressed genes (*ACTB*, *SNCA*, *CLCN2*, and *ORAI3*) in purified neutrophils from PD patients—provide novel insights into neutrophil regulation and function in PD.

Neutrophils are increasingly recognized for their phenotypic and functional diversity [44] in processes such as infection [45], inflammation [46], and neurodegeneration [47]. Their role in chronic degenerative diseases is only beginning to be understood. As the most abundant circulating leukocytes, neutrophils are readily accessible and offer great potential for biomarker discovery—especially through their membrane proteins, which are particularly suitable for clinical applications.

Interestingly, actin (*ACTB*) was initially included in our analysis as a reference gene.. Its upregulation in PD neutrophils was unexpected. As one of the most abundant proteins in eukaryotic cells, actin is involved in a wide array of protein–protein interactions [48], playing critical roles in cell motility, morphology, transcription, cytokinesis, and more [49]. In neutrophils, actin dynamics govern essential functions such as migration, cell division, and degranulation [50]. The cytoskeleton, comprising actin filaments, microtubules, intermediate filaments, and septins [51], is crucial for host defense mechanisms that rely on rapid cytoskeletal reorganization [52]. Cytoskeletal dysregulation is a known hallmark in Alzheimer’s disease, PD, and Huntington’s disease [53], with recent studies focusing on neuronal plasticity in PD [54]. However, its relevance in neutrophils during PD has not yet been explored. The observed actin upregulation suggests that neutrophil functions related to migration, phagocytosis, and degranulation may be altered in PD, warranting further investigation.

Alpha-synuclein (α-Syn) is a 140-amino-acid protein expressed in both the central nervous system and peripheral tissues, including skeletal muscle [55]. It participates in numerous cellular functions, such as phospholipase D inhibition [56], the regulation of synaptic vesicle trafficking [57], interactions with SNARE proteins [57], the modulation of tyrosine hydroxylase activity [58], and the binding to DJ-1 [59], synphilin [60,61], and tubulin [62,63]. It also enhances tau phosphorylation [64,65]. Pathological α-Syn aggregation occurs not only in PD but also in dementia with LBs, multiple system atrophy (MSA), and some forms of Alzheimer’s disease and traumatic brain injury. Although α-Syn expression in neutrophils has been previously reported, its specific role in these cells—and particularly its involvement in neurodegenerative pathways—remains unclear. Our findings suggest that α-Syn may contribute to organelle dysfunction in neutrophils, echoing its known roles in neuronal degeneration.

Calcium is a central intracellular messenger in non-excitable cells, including neutrophils. Store-operated calcium entry (SOCE) is the primary mechanism regulating intracellular calcium, linking endoplasmic reticulum (ER) depletion to calcium influx via calcium release-activated calcium (CRAC) channels. CRAC channels are formed by ORAI family proteins (ORAI1–3). In both mouse and human neutrophils, ORAI1 is the main functional component, though expression levels of ORAI isoforms can vary with inflammatory status. For example, resting bone marrow neutrophils in mice express more ORAI2, while inflamed joint neutrophils upregulate ORAI1 and ORAI3. CRAC channels are involved in regulating reactive oxygen species (ROS) production, chemotaxis, and phagocytosis. However, knockout models and patients with ORAI mutations do not show complete functional loss in neutrophils, suggesting redundancy among ORAI isoforms [66]. The observed upregulation of *ORAI3* in our study points to a possible shift in calcium signaling regulation in PD, which may underlie altered neutrophil functions.

Chloride channels (ClCs), encoded by the *CLC* gene family, form anion-selective pores that regulate membrane potential and cell volume [67]. ClC-2, one of nine mammalian ClCs [68], is a voltage-dependent, two-pore homodimeric channel [67] activated by membrane hyperpolarization, cell swelling, extracellular hypotonicity, or acidification [69,70,71]. It is widely expressed across human tissues and has been associated with diseases such as retinal and testicular degeneration, cystic fibrosis, leukoencephalopathy, epilepsy, and SBMA [32,33,34]. Despite its involvement in various degenerative disorders, the role of ClC-2 in PD—particularly in neutrophils—has not been previously explored.

Regarding our study population, we believe there are additional points worth discussing. The diagnosis of idiopathic PD was carried out by an experienced neurologist specializing in Movement Disorders, using a combination of physical examination, established clinical criteria, and standardized assessment tools. The mean disease duration of 9.75 years strongly reduces the likelihood of misdiagnosis with atypical Parkinsonian syndromes. Although the sample size was relatively small, the high number of technical replicates performed using dPCR compensates for the low *n*, thereby enhancing the robustness of our findings.

Another relevant consideration is the potential influence of anti-Parkinsonian medications on the immune response, particularly in neutrophils. To date, however, there is no evidence that the most commonly used treatments for Parkinson’s disease—levodopa, carbidopa, entacapone, rotigotine, rasagiline, and pramipexole—affect neutrophil function under chronic inflammatory conditions in humans. The only related finding is a study by Sadeghi et al. [72], which reported that pramipexole reduced neutrophil infiltration in a rat model of acute inflammation induced by carrageenan. Based on this, we consider it unlikely that our results were affected by pharmacological treatment; however, further studies specifically addressing this question are warranted to fully exclude any potential direct or indirect effects.

Altogether, the upregulation of *ACTB*, *SNCA*, *CLCN2*, and *ORAI3* supports the hypothesis that neutrophils in PD undergo functional changes related to migration, degranulation, ROS production, and phagocytosis. Discrepancies between theoretical and experimental gene regulation may be explained by differences in cell population: the GSE99039 dataset includes total leukocytes, whereas our study focused on neutrophil-enriched samples. While the direct role of neutrophils in PD pathophysiology remains to be elucidated, our results strongly support altered gene regulation in these cells and identify three novel membrane-associated biomarkers. These findings open new avenues for research on diagnostic, prognostic, and therapeutic strategies in Parkinson’s disease.

## 5. Conclusions

Our study provides theoretical evidence for the differential expression of 31 membrane protein-related genes in leukocytes from PD patients. These genes represent promising candidates for the identification of novel biomarkers in immune cells such as neutrophils and lymphocytes. Crucially, our findings underscore the need to investigate previously uncharacterized interactions among the proteins encoded by these genes to further clarify the role of leukocytes in PD pathophysiology. Experimentally, we report for the first time the expression and upregulation of the ClC-2 chloride channel in neutrophils from PD patients. Additionally, we demonstrate the upregulation of three other proteins—actin, alpha-synuclein , and ORAI3—suggesting functional alterations in neutrophils during PD, potentially affecting processes such as migration, phagocytosis, and degranulation. These results reinforce the contribution of inflammatory immune cells to PD and open new avenues for research focused on neutrophil biology, with potential implications for biomarker discovery and therapeutic targeting.

## Figures and Tables

**Figure 1 cimb-47-00459-f001:**
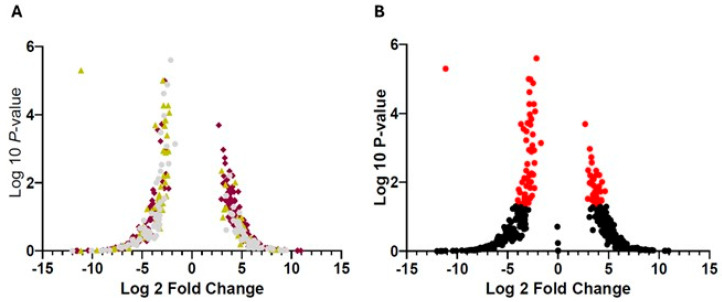
Regulation of genes from the GSE99039 database. Differentially expressed genes (DEGs) involved in immune function identified in whole blood samples from Parkinson’s disease (PD) patients. Volcano plots of DEGs in the GSE99039 dataset. Genes corresponding to voltage-regulated ion channels are shown in wine color, cytokine genes in yellow, and cell membrane receptor genes in gray. Upregulated genes (*p* > 0.05) are shown in red on the (**B**) plot, while downregulated genes (*p* ≤ 0.05) appear in red on the (**A**) plot. Non-significant genes are represented in black. The analysis was performed using SHINY GO version 0.81.

**Figure 2 cimb-47-00459-f002:**
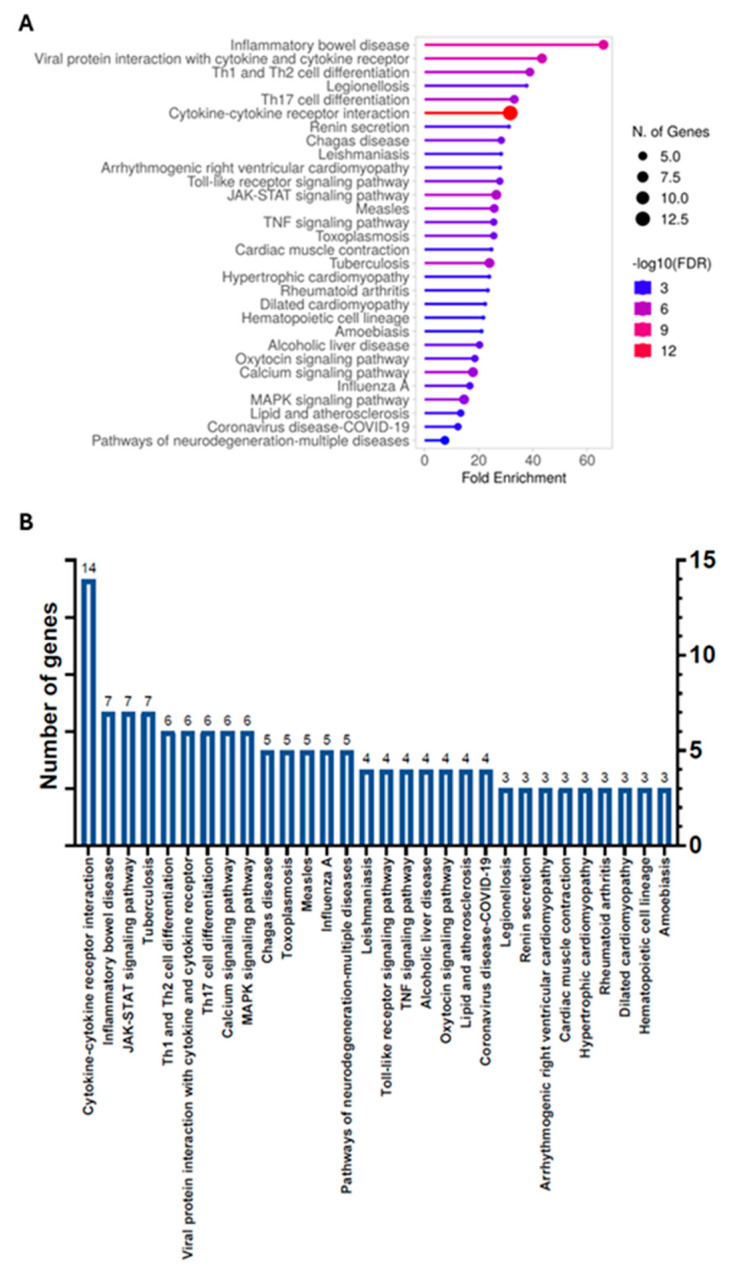
Enrichment pathway analysis. Gene enrichment analysis revealed that DEGs are predominantly associated with inflammatory processes. The “cytokine–cytokine receptor interaction”, “TLR”, and “TNF” signaling pathways exhibit a fold enrichment (FE) above 20, whereas the “neurodegeneration” pathway shows a lower FE. (**A**) FE of the top 30 enriched pathways. (**B**) Number of genes associated with each pathway.

**Figure 3 cimb-47-00459-f003:**
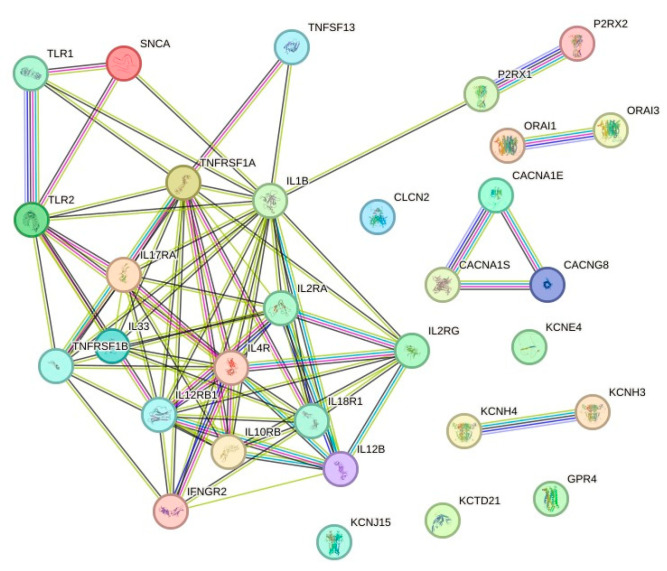
Protein–protein interaction (PPI) network analysis. The PPI network comprises 31 nodes and 81 edges, with an average local clustering coefficient of 0.705 and a PPI enrichment *p*-value of <1.0 × 10^−1^⁶. Edge colors indicate the type of interaction evidence: blue lines represent curated database entries; fuchsia, experimentally determined interactions; green, gene neighborhood; red, gene fusions; black, co-expression; and lilac, protein homology. The network was constructed using STRING software, version 12.0.

**Figure 4 cimb-47-00459-f004:**
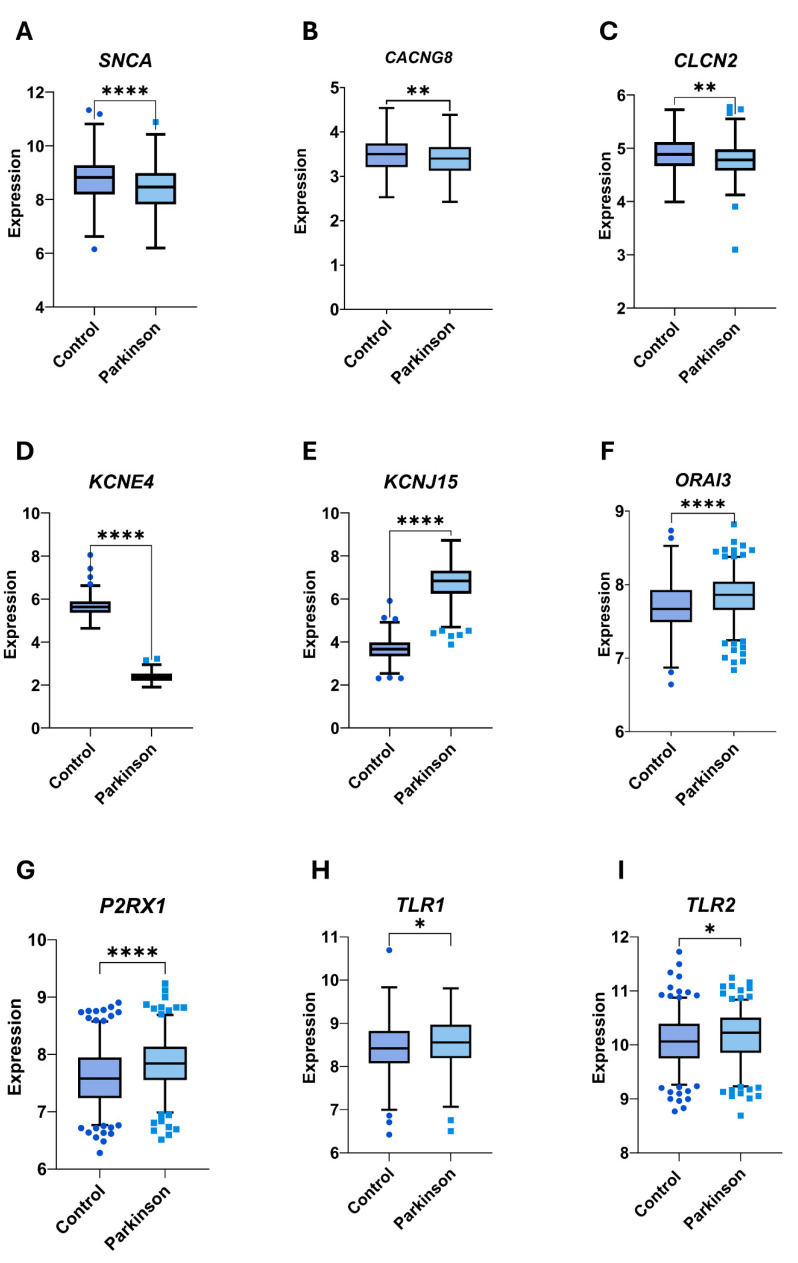
Graphical representation of differential expression of nine selected genes between PD patients and controls in data from the GSE99039 database. Each panel shows gene expression levels in fold changes after Log2 transformation on the *y*-axis and experimental groups on the *x*-axis (left: control; right: PD); * *p* ≤ 0.05; ** *p* ≤ 0.01; **** *p* ≤ 0.0001. (**A**) *SNCA* expression is significantly downregulated in the PD group compared to controls (*p* ≤ 0.0001). (**B**–**E**) Among the voltage-dependent ion channel genes, *CACNG8*, *CLCN2*, and *KCNE4* are significantly downregulated in PD (*p* ≤ 0.01, *p* ≤ 0.01, and *p* ≤ 0.0001, respectively), while *KCNJ15* is significantly upregulated (*p* ≤ 0.0001). (**F**–**I**) All four receptor genes (*TLR1, TLR2, ORAI3*, and *P2XR1*) show significantly upregulated expression in the PD group (*p* ≤ 0.05, *p* ≤ 0.05, *p* ≤ 0.0001, and *p* ≤ 0.0001, respectively). Statistical significance was determined using either Student’s *t*-test (*SNCA, CACNG8, ORAI3*, and *P2RX1*) or the Mann–Whitney *U* test (*CLCN2*, *KCNE4, KCNJ15*, *TLR1*, and *TLR2*).

**Figure 5 cimb-47-00459-f005:**
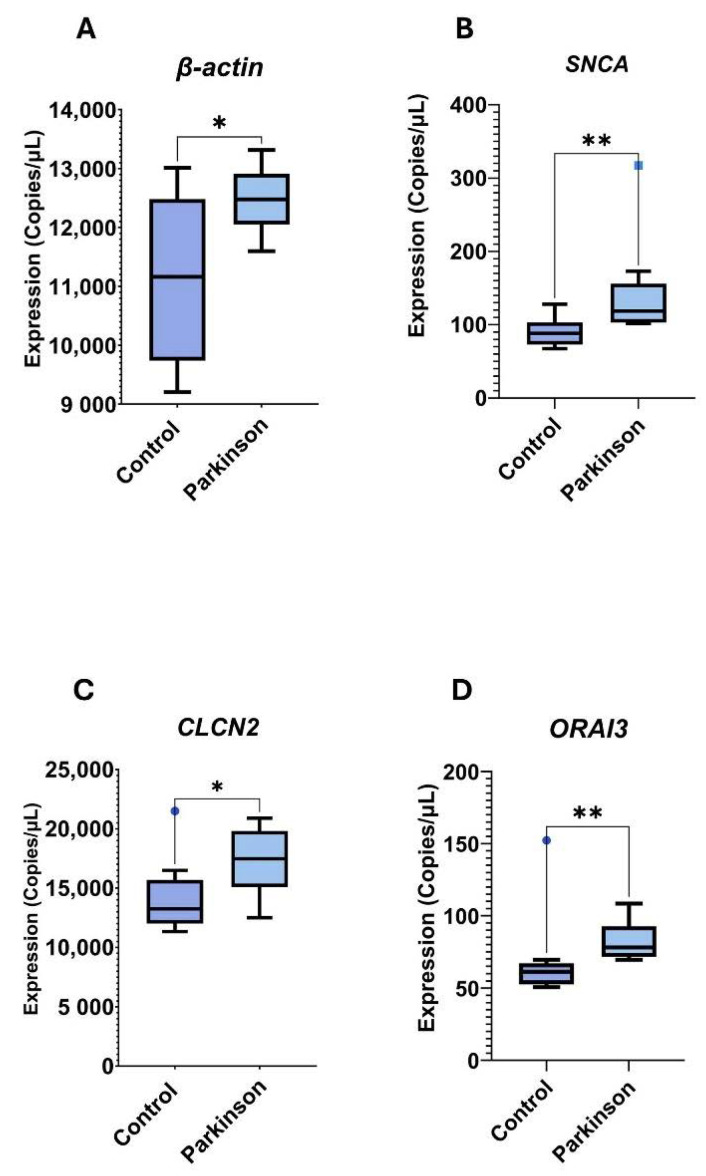
Gene expression levels analyzed by dPCR in neutrophils from PD patients and controls. Graphical representation of genes showing statistically significant differential expression; * *p* ≤ 0.05; ** *p* ≤ 0.01. (**A**) *ACTB* (β-actin) expression was significantly lower in the PD group compared to the control group (*p* ≤ 0.01). (**B**–**D**) Three genes—*SNCA*, *CLCN2*, and *ORAI3*—were significantly overexpressed in the PD group (*p* ≤ 0.001, *p* ≤ 0.01, and *p* ≤ 0.05, respectively). Statistical significance was determined using either Student’s *t*-test (*ACTB*) or the Mann–Whitney *U* test (*SNCA, CLCN2*, and *ORAI3*).

**Table 1 cimb-47-00459-t001:** List of genes showing significant statistical differences between PD and control groups in GSE99039 database.

#	ID	Symbol	SD	*p*-Value	FDR	Regulation
1	5023	* *P2RX1* *	−4.46 × 10^−1^	1.00 × 10^−4^	0.00000303	Up
2	7133	*TNFRSF1B*	−4.32 × 10^−1^	1.00 × 10^−4^	0.00000606	Up
3	93129	* *ORAI3* *	−4.09 × 10^−1^	1.00 × 10^−4^	0.00000909	Up
4	6622	* *SNCA* *	2.96 × 10^−1^	1.00 × 10^−4^	0.00003636	Up
5	3553	*IL1B*	−2.03 × 10^−1^	1.00 × 10^−4^	0.00005152	Up
6	23704	* *KCNE4* *	8.48 × 10^+00^	1.00 × 10^−4^	0.00005758	Down
7	3772	* *KCNJ15* *	−4.38 × 10^+00^	1.00 × 10^−4^	0.00007273	Up
8	3594	*IL12RB1*	−1.51 × 10^−1^	1.00 × 10^−4^	0.00008182	Up
9	3559	*IL2RA*	1.53 × 10^−1^	1.00 × 10^−4^	0.00008485	Down
10	59283	* *CACNG8* *	1.66 × 10^−1^	1.00 × 10^−4^	0.00008788	Down
11	777	*CACNA1E*	−6.74 × 10^−2^	1.00 × 10^−4^	0.00009091	Down
12	2828	*GPR4*	6.42 × 10^−1^	1.00 × 10^−4^	0.00009394	Down
13	23416	*KCNH3*	7.43 × 10^+00^	1.00 × 10^−4^	0.00009697	Down
14	23415	*KCNH4*	−2.90 × 10^+00^	1.00 × 10^−4^	0.00010000	Up
15	23765	*IL17RA*	−3.91 × 10^−1^	1.01 × 10^−4^	0.00001219	Up
16	7132	*TNFRSF1A*	−3.65 × 10^−1^	1.48 × 10^−4^	0.00002242	Up
17	3460	*IFNGR2*	−3.58 × 10^−1^	2.00 × 10^−4^	0.00003636	Up
18	3566	*IL4R*	−3.41 × 10^−1^	5.00 × 10^−4^	0.00010606	Up
19	8741	*TNFSF13*	−3.14 × 10^−1^	8.00 × 10^−4^	0.00019394	Up
20	22953	*P2RX4*	−8.03 × 10^−2^	4.80 × 10^−3^	0.00378182	Up
21	8809	*IL18R1*	−2.62 × 10^−1^	6.40 × 10^−3^	0.00193939	Up
22	90865	*IL33*	2.53 × 10^−1^	8.20 × 10^−3^	0.00273333	Down
23	3588	*IL10RB*	−3.23 × 10^−2^	1.72 × 10^−2^	0.00781818	Up
24	283219	*KCTD21*	−2.27 × 10^−1^	1.89 × 10^−2^	0.00801818	Up
25	1181	* *CLCN2* *	3.00 × 10^−1^	1.90 × 10^−2^	0.00518182	Down
26	7097	* *TLR2* *	−1.65 × 10^−1^	2.09 × 10^−2^	0.01456667	Up
27	779	*CACNA1S*	1.98 × 10^−1^	3.93 × 10^−2^	0.02143636	Down
28	7096	* *TLR1* *	−1.80 × 10^−1^	4.05 × 10^−2^	0.02700000	Up
29	3593	*IL12B*	1.56 × 10^−1^	4.13 × 10^−2^	0.03128788	Down
30	3561	*IL2RG*	3.32 × 10^−2^	4.13 × 10^−2^	0.02503030	Down
31	84876	*ORAI1*	−1.91 × 10^−1^	4.71 × 10^−2^	0.02997273	Up

Gray highlighted genes were chosen for a dPCR *p*-value ≤ 0.05. Abbreviations: SD: Standard Deviation; FDR: False Discovery Rate.

**Table 2 cimb-47-00459-t002:** Demographic characteristics of blood sample donors.

	Control (n = 9)	PD (n = 9)
Sex Men %(n) Women %(n)	66.7% (6) 33.3% (3)	55.5% (5) 44.5% (4)
Age (Mean ± SD)	63.92 ± 7.73	65.42 ± 9.75
High Blood Pressure Diagnosis %(n)	8.33% (1)	16.66% (2)
Years of PD diagnosis (Mean ± SD)	--	9.75 ± 6.38

**Table 3 cimb-47-00459-t003:** Statistical analysis of copies/uL of the analyzed genes by dPCR.

Gene	Control Copies/uL (Mean ± SD)	PD Copies/uL (Mean ± SD)	*p*
*SNCA*	90.16 ± 20.13	144.5 ± 68.9	0.0040
*ORAI3*	69.38 ± 31.71	81.7 ± 14.56	0.0044
*Actin*	11039 ± 1449	12511 ± 573	0.0120
*CLCN2*	14233 ± 3151	17429 ± 2821	0.0400
*P2RX1*	26.79 ± 4.748	22.77 ± 6.007	0.1135
*TLR2*	1326 ± 440.3	1641 ± 361.7	0.1172
*KCNE4*	713.4 ± 58.1	743.6 ± 55.53	0.2763
*TLR1*	2503 ± 1555	2997 ± 1220	0.4649
*CACNG8*	24.86 ± 4.102	25.04 ± 6.521	0.9436
*KCNJ15*	57.11 ± 16.11	56.67 ± 7.811	0.9641

## Data Availability

Data is contained within the article and Appendix A.

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
