# Peer review of "A Theoretical and Practical Analysis of Membrane Protein Genes Altered in Neutrophils in Parkinson’s Disease"

_cimb, 2025, doi:10.3390/cimb47060459_

Round 1
Reviewer 1 Report
Comments and Suggestions for Authors
The authors evaluated the fluctuation of membrane protein genes in neutrophils in patients with Parkinson's disease and showed that some proteins and genes were altered. This research was interesting, and it is considered that the findings obtained in this study could lead to the discovery of new marker molecules in PD, which could contribute to therapeutic interventions for PD patients.
However, the reviewer found some concerns, which are described below: I would appreciate the author's consideration.
First, fluctuations in actin levels are likely to indicate that the validation of experimental results should be questioned. Actin is a protein that exists constitutively and, as the authors state, it is a marker that is commonly used as a control because it does not fluctuate. If the authors wanted to demonstrate that actin is indeed altered by disease, we would need to select and evaluate an appropriate control protein other than actin and clearly show as the experimental results that the protein level of the control does not change.
At the same time, failure to select an appropriate control protein not only means that fluctuations in actin cannot be assessed but also that all other results on the levels of proteins and genes cannot be validated.
It seems that there is no quantitative correlation between the results on the expression level in Figure 4 and the dPCR results in Table 3. Please explain the reasons for this.
Why were the results analyzed using the t-test for significance? For example, the analytical results of the expression levels in Figure 4 show that even values with almost no difference have clear significant differences. Please consider whether appropriate statistical processing was performed.
Author Response
Reviewer 1:
We sincerely thank the reviewer for their thorough reading of our manuscript and for the thoughtful and constructive feedback provided. These comments have been invaluable in improving the quality and clarity of our work.
Below, we present our point-by-point responses to each of the reviewers’ comments.
1) REVIEWER: First, fluctuations in actin levels are likely to indicate that the validation of experimental results should be questioned. Actin is a protein that exists constitutively and, as the authors state, it is a marker that is commonly used as a control because it does not fluctuate. If the authors wanted to demonstrate that actin is indeed altered by disease, we would need to select and evaluate an appropriate control protein other than actin and clearly show as the experimental results that the protein level of the control does not change.
At the same time, failure to select an appropriate control protein not only means that fluctuations in actin cannot be assessed but also that all other results on the levels of proteins and genes cannot be validated.
RESPONSE: Actin was indeed intended as an internal control to demonstrate the expression of a constitutively expressed gene. However, we believe that its expression levels are not required to validate our experimental results, given that digital PCR (dPCR) is a highly sensitive and robust quantitative technique that enables absolute quantification without the need for a standard curve.
A key feature of dPCR is the partitioning of the PCR reaction mixture into thousands—or even millions—of individual micro-reactions. This approach allows the sample to be diluted to a level where each partition ideally contains either zero or one copy of the target nucleic acid molecule. While a small number of partitions may contain more than one molecule, the vast majority will contain either zero or one, particularly when working with low-concentration samples. As a result, the quantification of each gene is independent of a reference or control gene (1). One of the strengths of dPCR lies in the fact that direct and precise measurement of the target gene, independent of the amplification efficiency or standard curves. This technical advantage supports the robustness of our reported values, even in the absence of a stable endogenous control. Even more, the technique is used to validate reference genes (2).
After observing actin upregulation in the samples from Parkinson’s Disease (PD) patients, we recognized that actin is indeed essential for multiple neutrophil functions. This not only supports the relevance of neutrophil involvement and their differential regulation in PD, but also reinforces the role of membrane proteins in the disease, as actin plays a critical role in membrane organization and function.
For these experiments, the internal control used was the No Template Control (NTC), which the authors consider is enough to validate the technique.
2) REVIEWER: It seems that there is no quantitative correlation between the results on the expression level in Figure 4 and the dPCR results in Table 3. Please explain the reasons for this.
RESPONSE: Lines 276–278 state: “Figure 4 displays the statistical differences in gene expression between groups, as identified in the GSE99039 database, along with their corresponding p-values.”
Upon review, we realized that the original figure legend for Figure 4 did not explicitly indicate that the data were derived from the GSE99039 database. We have now added this information to the figure legend (lines 289–290) to ensure greater clarity.
Additionally, we have specified both in the main text and in the figure legend that gene expression levels are expressed as fold change values following Log2 transformation.
Regarding Table 3, it is directly related to Figure 5, as the gene expression levels presented in both formats correlate perfectly.
3) REVIEWER: Why were the results analyzed using the t-test for significance? For example, the analytical results of the expression levels in Figure 4 show that even values with almost no difference have clear significant differences. Please consider whether appropriate statistical processing was performed.
RESPONSE: We appreciate the reviewer’s thoughtful observation. In response, we have reviewed all statistical analyses and clarified the methods used to assess differences between groups. Specifically:
- We now explicitly state in the revised figure legends for Figures 4 and 5 which statistical test (Student’s t-test or Mann–Whitney U test) was used for each gene.
- The Statistical Analyses section of the manuscript has been updated to indicate that the Shapiro–Wilk test was used to evaluate normality. Based on these results, we applied the t-test for normally distributed variables and the Mann–Whitney U test otherwise.
- All statistical comparisons have been re-run to ensure consistency, and we confirm that the conclusions remain unchanged.
We agree that specifying the test used is essential for clarity and reproducibility. We are grateful for the reviewer’s insight, which led to this important clarification in the manuscript.
References
1.-Dube, S., Qin, J., & Rowe, L. A. (2008). Quantitation of DNA templates using Droplet Digital PCR. Nucleic Acids Research, 36(16), e125.
2.-Villegas-Ruíz, V.; Olmos-Valdez, K.; Castro-López, K.A.; Saucedo-Tepanecatl, V.E.; Ramírez-Chiquito, J.C.; Pérez-López, E.I.; Medina-Vera, I.; Juárez-Méndez, S. Identification and Validation of Novel Reference Genes in Acute Lymphoblastic Leukemia for Droplet Digital PCR. Genes 2019, 10, 376. https://doi.org/10.3390/genes10050376
Reviewer 2 Report
Comments and Suggestions for Authors
Re.: cimb-3659082
This is an interesting paper on Parkinson’s disease (PD) and possible blood-based biomarkers. The authors tested various genes in the neutrophils from blood samples and found increased expression (upregulation) of ORAI3, CLCN2, SNCA and β-actin in PD vs controls.
These results may prove important in our understanding of the interaction between ion channels, inflammatory and other membrane proteins in neutrophils of PD patients, as well as the processes of phagocytosis, degranulation and oxidative stress and how these mechanisms are associated with changes in the brain. Besides the above theoretical implications, neutrophils may offer easily accessible candidate biomarkers, which could be potentially helpful in clinical practice.
The introduction is excellent, presenting all aspects of PD. Methods are appropriate and statistics more than adequate. Results are well presented and the discussion is well written. Only 3 minor points:
(1) Subsection 2.6, line 139. How idiopathic PD was diagnosed (criteria?)? This is important in order to reduce inclusion of atypical parkinsonism mimicking PD.
(2) Since PD patients had a disease duration of at least 5y and received anti-Parkinson’s treatment, could the results of this study be due to treatment? Please acknowledge this possibility in the discussion section.
(3) In this paper PD was compared to controls only. Then, the question arises as to whether this upregulation is specific for PD or may be observed in other synucleinopathic or tauopathic atypical parkinsonism patients, in AD od FTD. This should be acknowledged in the conclusions, necessitating future studies.
Author Response
Reviewer 2
We sincerely appreciate the time and thoughtful comments provided by the reviewer. We are grateful for the opportunity to enhance the quality of our manuscript through this review process. In particular, we are thankful for the reviewer’s positive remarks and their recognition of the significance of our study.
In response to the recommendations, we provide our point-by-point replies below.
1) REVIEWER: Subsection 2.6, line 139. How idiopathic PD was diagnosed (criteria?)? This is important in order to reduce inclusion of atypical parkinsonism mimicking PD.
RESPONSE: All patients included in the study were diagnosed with idiopathic Parkinson’s disease by a certified neurologist, based on a combination of physical examination, established clinical criteria, and standardized assessment tools such as the UPDRS, MoCA, Hoehn & Yahr, and Epworth scales, as well as complementary studies, including electroencephalograms. This rigorous diagnostic process allowed us to exclude other forms of parkinsonism.
We also believe that the average disease duration of 9.75 years (as shown in Table 2) further supports the diagnosis, given that sustained response to treatment is consistent with idiopathic Parkinson’s disease. (Corresponding sentences were added in lines 418-432)
2) REVIEWER: Since PD patients had a disease duration of at least 5y and received anti-Parkinson’s treatment, could the results of this study be due to treatment? Please acknowledge this possibility in the discussion section.
RESPONSE: Due to the small sample size, we are currently unable to draw definitive correlations. However, as part of our exclusion criteria, none of the participants were taking anti-inflammatory medications, which could otherwise have influenced the results. While the impact of anti-Parkinson’s treatment was not among the primary objectives of this study, we acknowledge its potential relevance and will consider including this variable in future research designs.
The patients in our study were treated with the following anti-Parkinson’s medications: levodopa, carbidopa, entacapone, rotigotine, rasagiline, and pramipexole.
Upon reviewing the recent literature for evidence of anti-Parkinson’s treatments affecting neutrophil function in humans, we found no studies describing any effects of levodopa, carbidopa, entacapone, rotigotine, or rasagiline on the inflammatory process or on neutrophils specifically. Regarding pramipexole, a study by Sadeghi et al.. (1) reported an anti-inflammatory effect; however, this was observed in a rat model of acute inflammation induced by carrageenan or formalin. The authors demonstrated reduced neutrophil infiltration in the paws and ears.
While we agree that this finding warrants consideration and discussion, we believe it does not affect the interpretation of our results, as we did not evaluate neutrophil infiltration, and acute and chronic inflammation are distinct physiological processes. (Corresponding sentences were added in lines 418-432)
3) REVIEWER: In this paper PD was compared to controls only. Then, the question arises as to whether this upregulation is specific for PD or may be observed in other synucleinopathic or tauopathic atypical parkinsonism patients, in AD od FTD. This should be acknowledged in the conclusions, necessitating future studies.
RESPONSE: The findings obtained in this study have helped us establish a research line and have motivated us to further expand this work. In particular, we intend to incorporate comparisons with other neurodegenerative diseases in future studies. Both points (2 and 3) will be addressed in the Conclusion section of the revised manuscript, as they represent important considerations for future research directions.
We sincerely thank the reviewers and editors for their time, insightful comments, and efforts to improve our manuscript. Should any aspect remain unclear or require further clarification, we would be pleased to address it promptly.
References
1.-Sadeghi, H., Parishani, M., Akbartabar, M.T., Ghavamzadeh, M., Barmak, M.J., Zarezade, V., Delaviz, H. & Sadeghi, H. 2017. Pramipexole reduces inflammation in the experimental animal models of inflammation. Immunopharmacology and Immunotoxicology, 39, 80–86. https://doi.org/10.1080/08923973.2017.1284230
Round 2
Reviewer 1 Report
Comments and Suggestions for Authors
I would like to thank the authors for responding diligently to the reviewers' concerns one by one. The authors' correspondence have addressed the reviewer's concerns, and I believe the quality of revised manuscript has improved by this revision.